# International Survey of High-Flow Nasal Therapy Use for Respiratory Failure in Adult Patients

**DOI:** 10.3390/jcm12123911

**Published:** 2023-06-08

**Authors:** Asem Alnajada, Bronagh Blackwood, Ben Messer, Ivan Pavlov, Murali Shyamsundar

**Affiliations:** 1Wellcome Wolfson Institute for Experimental Medicine, Queen’s University Belfast, Belfast BT7 1NN, UK; aalnajada01@qub.ac.uk (A.A.);; 2Prince Sultan bin Abdulaziz College for Emergency Medical Services, King Saud University, Riyadh 11362, Saudi Arabia; 3The North East Assisted Ventilation Service, Royal Victoria Infirmary, Newcastle NE14LP, UK; 4Department of Emergency Medicine, Hôpital de Verdun, Montréal, QC H4G 2A3, Canada; ivan.pavlov.md@gmail.com; 5Regional Intensive Care, Royal Victoria Hospital, Belfast BT12 6BA, UK

**Keywords:** high-flow nasal cannula, acute hypoxemic respiratory failure, acute hypercapnic respiratory failure, chronic respiratory failure, acute settings, chronic settings

## Abstract

(1) Background: High-flow nasal therapy (HFNT) has shown several benefits in addressing respiratory failure. However, the quality of evidence and the guidance for safe practice are lacking. This survey aimed to understand HFNT practice and the needs of the clinical community to support safe practice. (2) Method: A survey questionnaire was developed and distributed to relevant healthcare professionals through national networks in the UK, USA and Canada; responses were collected between October 2020 and April 2021. (3) Results: In the UK and Canada, HFNT was used in 95% of hospitals, with the highest use being in the emergency department. HNFT was widely used outside of a critical care setting. HFNT was mostly used to treat acute type 1 respiratory failure (98%), followed by acute type 2 respiratory failure and chronic respiratory failure. Guideline development was felt to be important (96%) and urgent (81%). Auditing of practice was lacking in 71% of hospitals. In the USA, HFNT was broadly similar to UK and Canadian practice. (4) Conclusions: The survey results reveal several key points: (a) HFNT is used in clinical conditions with limited evidence; (b) there is a lack of auditing; (c) it is used in wards that may not have the appropriate skill mix; and (d) there is a lack of guidance for HFNT use.

## 1. Introduction

High-flow nasal oxygen therapy (HFNT) delivers a constant and precisely controlled blend of heated and humidified oxygen-air at flow rates of up to 60 L/min [1]. The putative benefits of HFNT are wide-ranging and apply to patients with both type 1 and type 2 respiratory failure. First, HFNT delivers a constant fraction of inspired oxygen with a high flow rate, matching the high inspiratory flow of patients with respiratory distress, which can reach up to 100 L/min [2]. Second, HFNT reduces anatomical dead space [3] by washing out expired air in the dead space. HFNT reduces rebreathing of carbon dioxide (CO_2_), which enhances ventilation and reduces partial pressures of carbon dioxide (PaCO_2_) and the work of breathing [4,5]. Finally, the provision of warm, humidified gas helps to avoid the drying up of secretions and preserves ciliary function that facilitates mucous clearance. There is some evidence that suggests HFNT improves comfort and is better tolerated than non-invasive ventilation (NIV) [6]. There are also fewer side effects, such as nasal and throat dryness and/or pain, associated with HFNT. This results in fewer episodes of dislodgement of the interface, increased therapy compliance and patient desaturation [1].

HFNT has been increasingly used in the current practice, especially for respiratory failure. Type 1 respiratory failure and type 2 respiratory failure can be managed by drug optimisation and by the delivery of oxygen through a nasal cannula or facemask [7,8]. Additional ventilatory support, such as NIV or invasive ventilation, is often required in case of further deterioration [8,9]. In acute type 1 respiratory failure (AT1RF), HFNT, as compared to NIV, may reduce intubation, mortality and hospital-acquired pneumonia and improve patients’ comfort, although the level of evidence is low [10]. Frat et al. [5] demonstrated that HFNT significantly reduced mortality rates in AT1RF at 90 days compared with NIV and conventional oxygen therapy (COT). The same study demonstrated that HFNT significantly reduced intubation rates in comparison to NIV and COT [5]. The above findings were supported by a systematic review investigating the role of HFNT in the intensive care unit (ICU), where HFNT is associated with a lower incidence of pneumonia and improved oxygenation [11]. However, in this systematic review, there was no difference in intubation and ICU length of stay. A recent systematic review comparing HFNT with COT for COVID-19 patients with acute respiratory failure demonstrated that HFNT significantly reduced intubation rates and mortality in the ICU compared to COT [12]. A randomised controlled trial (RCT) that included 604 patients and was conducted in three ICUs comparing HFNT with NIV for ICU patients with a high risk of reintubation showed non-inferiority of HFNT in preventing reintubation and post-extubation respiratory failure [13]. Another RCT conducted on 830 patients who underwent cardiothoracic surgery with a high risk of respiratory failure post-extubation showed non-inferiority of HFNT in treatment failure and ICU mortality when compared to NIV [14].

In patients with a high risk of reintubation, NIV is recommended to prevent reintubation, but studies have shown that NIV is associated with intolerance and discomfort [13,14,15], while HFNT is better tolerated [15]. After 12 h post-extubation, HFNT has been shown to reduce post-extubation respiratory failure and reintubation rates when compared to low-flow oxygen [13]. 

Currently, NIV is recommended for acute type 2 respiratory failure (AT2RF), and studies have shown that NIV can prevent intubation, reduce hospital and ICU stays and reduce mortality [9,16,17]. However, NIV has shown a high failure rate of up to 40–60%, which could lead to a delay in the initiation of invasive mechanical ventilation and mortality. There has been an increase in the use of HFNT for AT2RF [1]. Various observational studies have demonstrated that HFNT is beneficial in AT2RF, including improving gas exchange, acidosis, respiratory rate and work of breathing [18,19]. Small RCTs have demonstrated that HFNT is comparable to NIV in improving respiratory parameters (PaCO_2_, partial pressure of oxygen and pH) and patient-centred outcomes (incubation rate, mortality, length of hospital stay and patient comfort); this suggests non-inferiority of HFNT when compared to NIV [20,21,22,23]. A non-inferiority RCT conducted by Cortegiani et al. [24] compared HFNT to NIV as an initial respiratory support strategy for patients with acute exacerbation of chronic obstructive pulmonary disease (AECOPD) to investigate CO_2_ clearance after 2 h of treatment. The study showed a statistical non-inferiority of HFNT when compared to NIV in clearing CO_2_ after 2 h of treatment [24]. There are still no large multicentre clinical trials powered to detect clinical outcomes comparing HFNT with NIV.

In light of the potential benefits of HFNT as a new oxygen delivery method, its use is constantly expanding to address a variety of indications. While there are some guidelines that address the use of HFNT, these guidelines are not comprehensive [25,26]. It is imperative that we understand the current practice so that we may safely incorporate new technologies such as HFNT into routine practice. Despite the lack of solid evidence and scarcity of guidelines, the scope of HFNT use in clinical practice has been steadily expanding, and there is an urgent need to map current practice patterns to identify the need for future trials and guidelines. We designed a survey to understand current practices: the indications, the sites of use, the availability of local guidance, the availability of regular audits and the perceived need for better evidence or national guidelines.

## 2. Materials and Methods

An anonymous online survey was developed and prepared in Survey Monkey (Survey Monkey Inc., San Mateo, CA, USA). Respiratory therapists (RTs), respiratory physicians and emergency physicians were targeted as survey respondents due to their familiarity with acute respiratory failure patients across multiple locations and departments.

In the UK, the survey was distributed through the Association of Chartered Physiotherapists in Respiratory Care and the British Thoracic Society (BTS). In Canada, the survey was distributed through the Association of Emergency Medicine Specialists of Quebec, the Professional Order of Respiratory Therapists and the Quebec Association of Emergency Physicians. Furthermore, respiratory therapists responsible for device acquisition in each Canadian hospital were contacted through a list maintained by the Canadian division of Fisher & Paykel Healthcare Ltd., who had no involvement in the design of the survey, data collection, data analysis or manuscript preparation. Finally, the survey was also distributed in the USA through the American College of Emergency Medicine Physicians Research Group and the American Association of Respiratory Therapists. The survey responses were collected between October 2020 and April 2021. To encourage survey participation and maximise response rate, reminders were circulated at regular intervals via email or Twitter until the surveys closed. 

In the UK, we defined the institution denominator as the number of institutions listed within the British Thoracic Society (England, Scotland, Wales and Northern Ireland, *n* = 241). In Canada, the denominator was the number of institutions equipped with HFNT (*n* = 448). No denominator was available for institutions in the USA, and these data are presented separately. The UK version of the survey is provided in Appendix A.

## 3. Results

### 3.1. The UK and Canada Survey Results

There were 350 responses from the UK (165) and Canada (185). A total of 140 respondents did not provide their hospital affiliation, and the remaining respondents represented 25% (59/241) of UK and 24% (108/448) of Canadian hospitals. In the survey, the majority of respondents were RTs and physiotherapists (173/332; 52%), followed by consultants (112/332; 34%). Responses indicated that HFNT was used in 95% (333/350) of hospitals, with 31% (103/333) stating it was used in all wards. The most common areas were the Emergency Department (215/333; 65%), respiratory units (203/333; 61%) and medical units (185/333; 55%, Figure 1). Most clinicians used HFNT to treat AT1RF (327/333; 98%) followed by AT2RF (134/333; 40%) and in community chronic respiratory failure (CRF) (83/333; 25%, Figure 2). Community use of HFNT was significantly more prevalent in Canada (62/172; 36%) than in the UK (21/161; 13%, Table 1). Less than half of the respondents’ hospitals do not provide guidance on the usage of HFNT for AT1RF (135/328; 41%), while more than half of hospitals provide no guidance on HFNT usage for AT2RF (135/223; 61%, Figure 3) and with 79% (127/160) of hospitals provide no guidance on HFNT usage for CRF (Figure 3). The answers showed that the guidance provided by some hospitals mostly focused on when (102/333; 31%) and where (91/333; 27%) to start HFNT (Table 1).

HFNT indications are not regularly audited by hospitals, according to the majority of respondents (226/319; 71%). Additionally, 96% (310/324) of respondents considered the need for an HFNT guideline to be highly important, and 81% (208/258) believe that guidelines should be developed urgently (Table 1).

### 3.2. The USA Survey Results

The USA survey had 138 respondents, of which 70% (97/135) were respiratory therapists (Table 1). HFNT was used in 96% (133/138) of the respondents’ hospitals, with the highest use being in the emergency department (82%; 109/133, Table 1). HFNT was used for AT1RF (133/133; 100%), followed by AT2RF (64/133; 48%) and in the community (25/133; 19%, Table 1). Less than half of hospitals provide no guidance on the usage of HFNT for AT1RF (43/133; 32%), while more than half of hospitals provide no guidance on HFNT usage for AT2RF (37/87; 43%), and 47% (24/51) of hospitals do not provide guidance on HFNT usage for CRF (Table 1). Unlike the UK and Canada, the answers showed that the guidance focused on how to adjust FiO_2_ (65/133; 49%) and flow rate (68/133; 51%, Table 1). Broadly, in similar proportions to the UK/Canadian survey, the practice was poorly audited (79/131; 60%, Table 1). Additionally, 71% (92/131) of respondents considered the need for an HFNT guideline to be highly important, and 64% (55/86) believe guidelines should be developed urgently (Table 1). In the UK, 70% (91/130) of respondents indicated that they would be willing to participate in future HFNT RCTs (Table 1).

## 4. Discussion

This survey is the first of its kind to understand HFNT use and management in granular detail. It confirmed the widespread use of HFNT in different wards that are not the usual areas for respiratory support, such as surgical, orthopaedic and community settings. Further, it also identified the use of HFNT in clinical conditions where the quality of evidence is low such as AT2RF and in the community. 

Patients treated acutely with HFNT have significant mortality rates (11–17%) [27], and yet there is no mandated nursing ratio and no guidance for the location of care, unlike delivery of NIV. Lower nurse staffing and higher nurse workloads are related to adverse patient outcomes such as mortality, infections and longer hospital stay [5] and may compromise care provided to patients supported with HFNT outside of specialised units. There is a lack of practical HFNT guidelines from medical societies. The European Society of Intensive Care Medicine has produced a practice guideline with a focus on hypoxaemic respiratory failure, peri-intubation, post-extubation and peri-operative care [26]. The guideline does not provide any recommendation on staffing ratio, monitoring, escalation or de-escalation guidance. The BTS guideline for oxygen use in adults and emergency settings mentions HFNT briefly as an option for patients with AT1RF without any additional guidance [28].

In our survey, a significant proportion of hospitals deliver HFNT in hospital wards that may not be suited to manage these patients. Close monitoring of these patients to identify deterioration is paramount, as delayed intubation may result in increased mortality [29]. Incorporating HFNT in early warning scores would enable the identification of high-risk patients. Delayed HFNT weaning can be associated with an unnecessary increase in hospital stays with ramifications on healthcare-associated costs, as shown in the paediatric population [30]. The lack of a governance structure relating to HFNT contrasts with that of NIV, which has a better evidence base and is subject to national audits and quality standards [31]. 

The survey highlighted the use of HFNT for clinical conditions where the quality of evidence is poor, such as AT2RF [4], which may not be cost-effective, if not harmful for some patients by delaying NIV or intubation. Moreover, this increased uptake of HNFT increases the difficulty of generating high-quality randomised evidence comparing HFNT to other oxygenation modalities. There is an urgent need for HFNT-specific guidelines that focus on safe delivery, identification of therapy response and clinical situations beyond AT1RF. The guidelines should identify key research gaps to guide research priorities.

Clinical auditing is key to improving the quality of patient care collaboratively and systematically. This survey has demonstrated the lack of this key patient safety measure in HFNT practice.

The strengths of this survey are the completion of the survey by clinicians from three countries, with representation from more than a quarter of UK and Canadian hospitals, which suggests the generalisability and validity of the survey despite the relatively low number of individual responses. Similar responses to the survey in the UK and North America demonstrate that the issues surrounding HFNT delivery are widespread. The main limitation of the survey is the low number of individual responses and uncertainty of the number of units.

## 5. Conclusions

In conclusion, HFNT is utilised in multiple areas in the hospital for clinical conditions where the evidence is poor or lacking, and its practice is not widely audited. The development of practical guidelines was felt to be important and urgent by over 80% of respondents. Most respondents felt that there is a need for further trials of HFNT in common causes of respiratory failure.

## Figures and Tables

**Figure 1 jcm-12-03911-f001:**
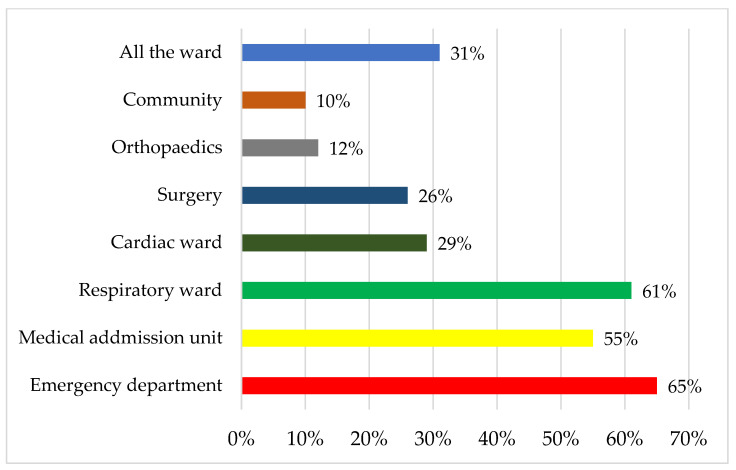
Location of high flow nasal therapy use in the UK and Canada.

**Figure 2 jcm-12-03911-f002:**
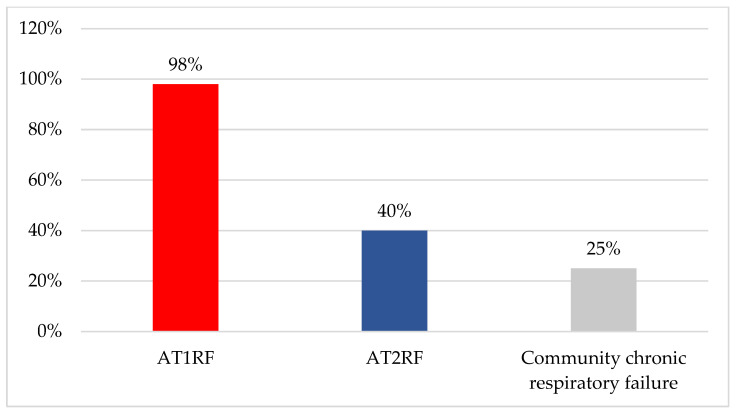
The most common conditions treated with HFNT in the UK and Canada. Abbreviations: AT1RF, acute type 1 respiratory failure; AT2RF, acute type 2 respiratory failure.

**Figure 3 jcm-12-03911-f003:**
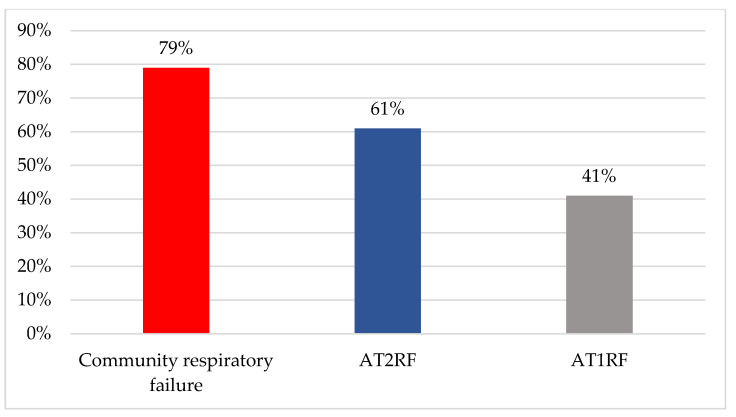
The lack of high flow nasal therapy guidance in the UK and Canada. Abbreviations: AT1RF, acute type 1 respiratory failure; AT2RF, acute type 2 respiratory failure.

**Table 1 jcm-12-03911-t001:** Detailed survey results of all sites.

Survey Items		UK n/N (%)	USA n/N (%)	Canada n/N (%)
HFNT used in any wards?	Yes	161/165 (98%)	133/138 (96%)	172/185 (93%)
	no	4/165 (6.1%)	5/138 (4%)	13/185 (7%)
Respondents’ professions	Consultants	66/161 (41%)	36/133 (26%)	46/171 (27%)
	Physiotherapists/RT	51/161 (32%)	97/133 (73%)	122/171 (71%)
	Nurse	16/161 (10%)	0/133 (0%)	2/171 (1%)
	Trainee	23/161 (14%)	2/133 (2%)	1/171 (1%)
	Other	5/161 (3%)	0/133 (0%)	0/171 (0%)
Location of use	Emergency department	58/161 (36%)	109/133 (82%)	161/175 (92%)
	Medical admission unit	67/161 (42%)	85/133 (64%)	118/175 (67%)
	Respiratory ward	132/161 (82%)	67/133 (50%)	157/172 (91%)
	Cardiac ward	40/161 (25%)	61/133 (46%)	117/172 (68%)
	Surgery	34/161 (21%)	29/133 (22%)	71/172 (41%)
	Orthopaedics	16/161 (10%)	27/133 (20%)	58/172 (34%)
	Community	8/161 (5%)	2/133 (2%)	54/172 (31%)
	All the wards	28/161 (17%)	58/133 (44%)	24/172 (14%)
Conditions treated with HFNT	AT1RF	159/161 (99%)	133/133 (100%)	168/172 (98%)
	AT2RF	41/161 (25%)	64/133 (48%)	93/172 (54%)
	CRF	21/161 (13%)	25/133 (19%)	62/172 (36%)
Guidance availability for AT1RF	Yes	65/158 (41%)	76/133 (57%)	57/170 (34%)
	No	42/158 (27%)	43/133 (32%)	93/170 (53%)
	Unsure	51/158 (32%)	14/133 (11%)	23/170 (14%)
Guidance availability for AT2RF	Yes	17/108 (16%)	35/87 (40%)	20/115 (17%)
	No	59/108 (55%)	37/87 (43%)	77/115 (67%)
	Unsure	32/108 (30%)	15/87 (17%)	18/115 (16%)
Guidance availability for CRF	Yes	2/70 (2%)	13/51 (25%)	5/90 (6%)
	No	66/70 (66%)	24/51 (47%)	61/90 (61%)
	Unsure	22/70 (31%)	14/51 (27%)	24/90 (27%)
Components of the guidelines	When to start HFNT	63/161 (39%)	52/133 (39%)	39/172 (23%)
	Where to start HFNT	57/161 (35%)	38/133 (29%)	34/172 (20%)
	HFNT is used to classify the level of patient care	14/161 (9%)	45/133 (34%)	23/172 (13%)
	Guidance on nursing ratios	12/161 (7%)	9/133 (7%)	10/172 (6%)
	Guidance on achieving training competency	24/161 (15%)	30/133 (23%)	8/172 (5%)
	How to monitor patients on HFNT?	44/161 (27%)	58/133 (44%)	40/172 (23%)
	How to adjust FiO_2_?	45/161 (28%)	65/133 (49%)	41/172 (24%)
	How to adjust the flow rate?	43/161 (27%)	68/133 (51%)	38/172 (22%)
	How to wean off HFNT?	40/161 (25%)	57/133 (43%)	35/172 (20%)
	Is there an escalation policy?	48/161 (30%)	24/133 (18%)	25/172 (15%)
Common indications for HFNT use in AT1RF	Acute bronchial asthma	39/161 (24%)	64/133 (48%)	78/172 (45%)
	Bronchiectasis	72/161 (45%)	54/133 (41%)	81/172 (45%)
	COPD	72/161 (45%)	98/133 (74%)	134/172 (76%)
	Cystic fibrosis	39/161 (24%)	0/133 (0%)	18/172 (10%)
	Interstitial lung disease	121/161 (73%)	92/133 (69%)	104/172 (60%)
	Obstructive sleep apnoea (without hypercapnia)	20/161 (12%)	27/133 (20%)	19/172 (11%)
	Neurological conditions	33/161 (20%)	17/133 (13%)	38/172 (22%)
	Pneumonia	135/161 (84%)	112/133 (84%)	150/172 (87%)
	Pulmonary embolism	63/161 (39%)	42/133 (32%)	56/172 (33%)
	Pulmonary oedema	74/161 (46%)	79/133 (59%)	85/172 (49%)
Common indications for HFNT use in AT2RF	AECOPD	40/161 (25%)	61/133 (46%)	88/172 (51%)
	Bronchiectasis	17/161 (11%)	32/133 (24%)	40/172 (23%)
	Cystic fibrosis	14/161 (9%)	18/133 (14%)	34/172 (20%)
	Drug overdose	13/161 (8%)	7/133 (5%)	23/172 (13%)
	Neurological conditions	18/161 (11%)	12/133 (9%)	21/172 (12%)
	Obesity hypoventilation syndrome	12/161 (7%)	33/133 (25%)	30/172 (17%)
Availability of HFNT regular audit	Yes	39/147 (27%)	16/131 (12%)	12/176 (7%)
	No	108/147 (73%)	79/131 (60%)	118/176 (69%)
	Unsure	N/A	36/131 (27%)	42/176 (24%)
Importance of having an official guideline for HFNT	Very important	98/160 (61%)	92/131 (71%)	97/164 (59%)
	Important	53/160 (33%)	29/131 (22%)	62/164 (38%)
	Not important	9/160 (6%)	10/131 (7%)	5/164 (3%)
The urgency of having an official guideline for HFNT use	Urgent	115/149 (77%)	76/86 (88%)	93/112 (83%)
	Not urgent	34/151 (23%)	10/86 (11%)	19/112 (17%)
Need for a trial of HFNT in the following conditions:	AT1RF due to community-acquired pneumonia	124/161 (77%)	91/133 (68%)	113/172 (66%)
	AT1RF due to hospital-acquired pneumonia	115/161 (71%)	86/133 (65%)	108/172 (62%)
	Acute severe asthma	55/161 (34%)	61/133 (46%)	101/172 (59%)
	AECOPD	70/161 (43%)	80/133 (60%)	108/172 (63%)
Taking part in the clinical trial for various conditions	Yes	91/130 (70%)	N/A	N/A
	No	39/130 (30%)	N/A	N/A

Abbreviations: AARC, American Association for Respiratory Care; ACPRC, Association of Chartered Physiotherapists in Respiratory Care; ACEMP, American College of Emergency Medicine physicians research group; AECOPD, acute exacerbation of chronic obstructive pulmonary disease; AMUQ, Association des médecins d’urgence du Québec (Quebec Association of Emergency Physicians); ASMUQ, Association des spécialistes en médecine d’urgence du Québec, (Quebec Association of Emergency Medicine Specialists); AT1RF, acute type 1 respiratory failure; AT2RF, acute type 2 respiratory failure; BTS, British Thoracic Society; CRF, chronic respiratory failure; FiO_2_, fractional inspired oxygen; HFNT, high-flow nasal therapy; N/n, number of respondents; N/A, not applicable; OPIQ, Ordre professionnel des inhalothérapeutes du Québec (Professional order of Quebec respiratory therapists); RT, respiratory therapists; UK, United Kingdom; USA, United States of America.

## Data Availability

Complete data are available in Table 1.

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
