# Peer review of "International Survey of High-Flow Nasal Therapy Use for Respiratory Failure in Adult Patients"

_jcm, 2023, doi:10.3390/jcm12123911_

Round 1

Reviewer 1 Report

The authors here present the results of a survey study on the use of HFNT in the UK, Canada, and the US and the perceived lack of institutional and professional association guidelines. 

1) The figures which are presented need to be improved. Typically, pie charts add up to 100%. Neither of these do which is confusing. Perhaps a simple bar graph (like figure 1) would be more useful.

2) Why did the authors lump physiotherapists and respiratory therapists in the same category? At least in the US these are rather different specialties with different clinical emphasis. Could this have caused either confusion in answering the questions or caused the data to be skewed (or represent neither side adequately)?

3) When asking if the respondents thought that the need for guidelines was important, the question answers are phrased as not important, important, and very important. Does this skew the answer choices (and the conclusions) in a more critical direction given that even the middle option implies a critical need? This would have been mitigated if there had been a 4th option or if the middle option had been removed.

Author Response

We thank the reviewer for the responses. Please see the attachment.

Reviewer 2 Report

The author designed a survey to understand current practices: the 96 indications, the sites of use, the availability of local guidance, the availability of regular audits, and the perceived need for better evidence or national guidelines. They found HFNT is utilised in multiple areas in the hospital, for clinical conditions where the evidence-based is poor or lacking, and its practice is not widely audited.  The development of practical guidelines was felt to be important and urgent by over 80%  Most respondents felt that there is a need for further trials of HFNT in common causes of respiratory failure. Fine study. 

Author Response

We thank the reviewer for the response. Please see the attachment.

Reviewer 3 Report

Review for the manuscript International Survey of High-Flow Nasal Therapy Use for Respiratory Failure in Adult Patients”.

The study designed a survey to understand current practices regarding the use of HFNT, indications, the sites of use, the availability of local guidance, the availability of regular audits, and the perceived need for better evidence or national guidelines. The survey was distributed in the UK, USA and Canada and the survey responses were collected between October 2020 and April 2021.

The survey suggested the use of HFNT for clinical conditions where the quality of evidence is poor such as AT2RF .

 Comments

 The study is very interesting and the results are important for clinicians.

The methods are adequately described.

The results are clearly presented and the conclussions supported by the results.

There are one table and three figures which sustained the results. 

Author Response

We would like to thank the reviewer for the response. Please see the attachment.
